# Current Clubfoot Practices: POSNA Membership Survey

**DOI:** 10.3390/children10030439

**Published:** 2023-02-23

**Authors:** Oliver C. Sax, Larysa P. Hlukha, John E. Herzenberg, Philip K. McClure

**Affiliations:** 1Center for Joint Preservation and Replacement, Rubin Institute of Advanced Orthopedics, Sinai Hospital of Baltimore, 2401 W. Belvedere Ave., Baltimore, MD 21215, USA; 2International Center for Limb Lengthening, Rubin Institute of Advanced Orthopedics, Sinai Hospital of Baltimore, 2401 W. Belvedere Ave., Baltimore, MD 21215, USA

**Keywords:** anesthesia, cast removal, plaster, Ponseti, tenotomy

## Abstract

Clubfoot management has advanced in the 21st century with increases in formal training, practitioner experience, and improved casting/bracing constructs. The Ponseti method is the gold standard, yet variations in application persist. This survey aims to identify current treatment practices among clubfoot practitioners within the Pediatric Orthopaedic Society of North America (POSNA). A 23-question online survey of members was conducted between June and August 2021. Eighty-nine respondents self-identified as clubfoot providers. Of these, 93.1% had an MD degree, 23.6% possessed >30 years’ experience, and the majority (65.6%) worked in a teaching hospital associated with a medical school. Most responders (92.0%) were pediatric fellowship trained. A total of 51.7% had participated in a clubfoot training course. More than half (57.5%) noted changes to clubfoot management practices throughout their training. A majority used between four and seven (88.7%) long leg casts (98.4%), changed at seven-day intervals (93.4%). Plaster (69.4%) was most commonly used. The most common bracing device was Mitchell–Ponseti (72.9%). A mean 84.8% of clubfeet required tenotomy. The most common anesthetic agent was numbing gel (43.0%). Tenotomies mostly occurred in patients aged <6 months (63.1%). Tenotomy locations were operating room (46.5%), clinic (45.4%) and procedure room (8.1%). Cast removal was primarily performed with saws (54.7%). The mean incidence of observed cast burns was 5.5%. Most providers did not use a device to prevent cast burns (76.6%). Reported cast complications included slippage (85.9%), skin irritation (75.8%), and saw-related injuries (35.9%). Clubfoot management variations exist in orthotics, tenotomy indications and practices, and cast material. Casting complications continue to be a problem. Further studies are warranted to determine if certain practices predispose patients to specific complications.

## 1. Introduction

Most pediatric orthopedic surgeons treating clubfoot utilize the Ponseti method [1,2,3]. The Ponseti method has been the gold standard of clubfoot treatment since its introduction in the late 1940s, with utilization of abduction bracing and tibialis anterior transfer for recalcitrant deformity [4]. In his original work, Dr. Ponseti corrected clubfoot deformity using gentle manipulation with well-molded and thinly padded plaster of Paris casts, reapplied every four to seven days. Subsequent teachings and refinements since the late 1990s have resulted in a variety of modalities available to current practitioners. For example, plaster of Paris was the traditional material used in the serial casting of patients; however, reports suggest comparable results when using semi-rigid fiberglass [5,6]. Achilles tenotomy also has seen an evolution in indications, the setting for the procedure, and anesthesia choice. Scoring systems are available to guide decision-making for tenotomies, but their indication profiles may not necessarily overlap in specific patients [7,8]. The Pirani and Diméglio classification systems are the two most widely used for clubfoot assessment [9,10]. Although they are unique in their approach, they have proven to be complementary to each other, with good correlation coefficients reported [11]. The evolution of clubfoot management continues since Dr. Ponseti’s original publication. The Ponseti method originally evolved as serial manipulations and castings performed in the clinic with tenotomy under numbing gel or local anesthesia. Several anesthesia choices for tenotomies are available to providers, especially given advances in pediatric anesthesia since the 1980s. Some report the use of light sedation involving mask induction with oxygen/nitrous oxide, while others opt to provide general anesthesia via intravenous propofol infusion and supplemental oxygen in the setting of an operating room (OR) [12,13,14,15].

Clubfoot is an idiopathic developmental deformation characterized by the excessive shortening of the tibialis posterior tendon resulting in the classic “CAVE” deformity (Cavus; forefoot Adductus; hindfoot Valgus; and Equinus contracture). It represents one of the most prevalent congenital birth defects with an estimated 1 per 1000 live births affected [16]. Although poorly understood, the etiology of clubfoot may be attributed to intra-uterine posture, with some literature suggesting alternatives related to mesodermal, musculoskeletal, neurologic, or vasculature mechanisms [17]. Environmental and genetic factors have been posited to contribute to the etiology of clubfoot, specifically with the expression of both *PITX1* and *TBX4* pathways and their role in hindlimb development [18]. Regardless of etiology, clubfoot has successfully been managed for several decades. Despite the method’s success, patients treated using the Ponseti technique have been shown to achieve delayed independent gait [19]. However, this may not be surprising considering the initial deformity that impedes the child’s ability to ambulate.

The reasons behind the evolution of clubfoot practices may be multifactorial, with advancements in anesthesia techniques, practitioner experience, varied payers, and modifications to the standard Ponseti method, amongst others. However, the degree of variation among current practitioners is not well understood. Several previous studies describe clubfoot survey results from the Pediatric Orthopaedic Society of North America (POSNA) membership. The first survey regarding clubfoot practices among POSNA members was conducted in 2003 by Heilig et al., who reported on the current trends of clubfoot management [20]. In 2012, Zionts et al. surveyed POSNA members to assess clubfoot treatment practices with a particular focus on treatment changes since a 2001 survey [1]. A 2019 survey by Hosseinzadeh et al. specifically assessed clubfoot relapse and reported providers’ treatment recommendations [21]. Despite the importance of the most recent surveys, they were limited to general discussion with some attention given to relapse. They did not explore specifics about casting materials and indications for tenotomies [15,21]. Therefore, the purpose of this study is to provide results from a current survey of clubfoot practices among POSNA members with a specific focus on examining: (1) demographic and educational characteristics of clubfoot practitioners; (2) casting preferences; (3) Achilles tenotomy indications and anesthesia preferences; and (4) cast removal technique preferences.

## 2. Materials and Methods

A 23-question survey was developed by the authors and approved by the POSNA Evidence Based Practice Committee. Our institutional review board reviewed this and determined it to be exempt. The survey was emailed to all POSNA members in August 2021. Members are pediatric orthopedic-trained practitioners and allied health professionals. Only members indicating they were clubfoot practitioners had access to the complete survey. Demographic information included professional degrees, years in practice, type of practice, location of practice, fellowship experience, practice changes, and prior participation in clubfoot courses. Casting information included the number of casts commonly used, the interval between castings, the type and material of cast, and bracing preferences. Achilles tenotomy indications and anesthesia preferences included the percentage of patients requiring tenotomy, the type of anesthesia used, the setting for the tenotomy, and the indications for bedside tenotomy. The cast removal variables included removal preferences, the percentage of cast burns observed, the use of skin protectors to prevent cast saw injury, and other cast problems observed.

Several survey questions provided multiple answer choices, whereupon the member had the ability to select one or more options. Answers were tallied and listed as a percentage of the total number of respondents. Other survey questions required respondents to slide a scale to indicate a percentage, such as the proportion of patients requiring tenotomy and the incidence of cast saw burns.

## 3. Results

### 3.1. Demographics and Education

At the time of this survey, we contacted the entire POSNA membership (n = 1223). Eighty-nine members (7.3%) responded that they were clubfoot providers (Table 1). The most common degrees were MD (93.3%), followed by DO (4.5%), PhD (3.4%), and other (2.3%). The largest group was in practice for >30 years (23.6%) and nearly all respondents were pediatric fellowship trained orthopedic surgeons (92.1%). The majority worked in a major teaching hospital with a medical school (65.2%) and practiced in a university-based system (44.2%). The most common locations of practice included Texas (11.9%), California (9.5%), and Canada (6.0%) (Table 2).

### 3.2. Clubfoot Training and Management

Most respondents indicated a change in clubfoot management over the course of their training (57.5%) (Table 3). Popular clubfoot training courses included: Iowa with Dr. Ponseti (36.4%), Iowa after Dr. Ponseti (27.3%), International Pediatric Orthopedic Symposium (59.1%), and Baltimore Limb Deformity Course seminars (31.8%). For the management of clubfoot, most utilize four to seven (88.7%) long leg casts (98.4%), for seven days’ duration (93.4%). Plaster (69.4%) and semi-rigid (27.4%) materials were predominantly used. Bracing choices offered to patients were most commonly Mitchell–Ponseti (72.9%), Denis Browne (60.8%), and Dobbs (31.7%) constructs.

### 3.3. Tenotomy and Anesthesia

Respondents indicated that a mean of 84.8% of patients required a tenotomy, with an age indication of <6 months (63.1%). The type of anesthesia varied, but most used local injection (39.5%) and/or numbing gel (43.0%). Sedation was less commonly utilized; 7% reported the use of conscious sedation and 33.7% general anesthesia. The tenotomy venue was evenly split between the OR (46.5%) and the clinic/office (45.4%), while others performed it in a procedure room (8.1%).

### 3.4. Clubfoot Outcomes and Complications

If a plaster cast was employed, most respondents indicated the oscillating cast saw as their choice of removal mechanism (54.7%), while others soaked it off in water either at home (11.6%) or in clinic (9.3%). The mean incidence of cast saw burns was 5.5%. Cast saw burn injuries were acknowledged to have been witnessed by 35.9% of those surveyed. Most providers indicated not using any device to prevent this injury (76.6%) (e.g., plastic strips, woven tape, et cetera). Other cast problems observed included cast slippage (85.9%) and skin irritation (75.8%).

## 4. Discussion

The Ponseti method remains the gold standard management option for clubfoot [1,2,22,23,24]. However, the continuous advancement and subspecialization of pediatric orthopedics over time have resulted in practice changes among clubfoot providers. First, the number of clubfoot providers is relatively small. Of 1223 POSNA members surveyed, only 89 self-identified as clubfoot providers and completed the entire survey. This is not surprising, given the trend towards subspecialization. In typical major university programs or free-standing children’s hospitals with 10–20 pediatric orthopedic attendings, only 1–2 will be concentrating on clubfoot. Thus, if only about 10% of pediatric orthopedists are treating clubfoot regularly, the number of POSNA members fitting this description would be about 122. Therefore, rather than a low response rate, we believe our survey likely represents a true picture of the state of clubfoot management.

To capture an updated understanding amongst current clubfoot providers, we surveyed the POSNA membership in 2021. We found a decreased utilization of plaster of Paris compared with prior surveys, and an increased use of general anesthesia in the OR compared with prior surveys. Although further surveys are needed to draw definitive conclusions, our data shed light on a possible transition in clubfoot practices. There may be a trend of fewer, older, more experienced pediatric orthopedic surgeons treating clubfeet.

The Ponseti method describes plaster cast applications followed by a bracing phase. A 2012 clubfoot survey [1] demonstrated plaster usage among 70.2% of providers, similar to the 69.4% in this present study. Our respondents indicate a trend towards alternative casting materials such as semi-rigid casts. This is consistent with an overall declining preference for the plaster of Paris in favor of synthetic casting, though the former continues to be widely used. Several studies highlight the efficacy of synthetic casts including lower rates of cast slippage and soft tissue injury. In a retrospective comparative study of synthetic versus plaster of Paris in 136 clubfeet, Monforte et al. [5] found similar cases of relapse without complications in either group. However, a shorter duration of cast use was reported in those receiving synthetic casts, possibly indicating a more mildly affected group. A similar study was conducted by Pittner et al. [6] to determine differences between plaster and flexible fiberglass casts in 42 clubfeet. Using the Diméglio scoring system [10] for clubfoot deformity severity, Pittner et al. [6] found higher mean post-cast scores among flexible fiberglass recipients (6.4 versus 4.1, *p* = 0.037), despite similar initial scores, lending evidence to the possibility of decreased effectiveness. However, a statistically insignificant trend toward higher patient satisfaction was noted in the flexible fiberglass-treated recipients [6]. In a randomized controlled trial of 133 clubfeet treated with either plaster of Paris or semi-rigid synthetic casts, Aydin et al. [25] found no differences in the number of casts applied until tenotomy, but higher cast slippage and cast/skin lesions among plaster recipients. Although the studies do not point to a recognizable superiority of synthetic casts when compared with plaster of Paris, their ease of removal may account for their acceptance as an alternative to plaster.

Traditional tenotomy management for clubfoot was comprised of the minimal use of anesthesia, with some utilizing gel or local anesthetic performed in a clinic room. Despite this traditional recommendation, a 2012 study by Zionts et al. [1] reported a large proportion of systemic anesthesia use for Achilles tenotomy. Local anesthesia was used for 39.4%, while conscious and general sedation was administered in 7.3 and 45.4%, respectively. We interpret our results to be similar, with a small inclination away from the OR: local anesthetic (39.5%) or numbing gel (43.0%), with some providers utilizing conscious (7.0%) and general sedation (33.7%).

Regarding anesthesia usage, multiple studies highlight the role of sedation in clubfoot management. In a study of 114 patients, Iravani et al. [15] demonstrated the safety of tenotomy under propofol sedation. Bor et al. [12] compared twenty-four patients who underwent conscious sedation via oxygen/nitrous oxide with five patients who underwent general sedation via propofol infusion. No complications were reported, and light sedation was recommended when encountering older patients who may struggle with local anesthesia alone. However, performing a tenotomy as an office procedure is reportedly safe and effective. Lebel et al. [14] retrospectively reviewed 56 babies who underwent tenotomy without use of sedation or general anesthesia. They reported no adverse events related to local anesthesia, inferring that tenotomy without sedation can be safely performed. The cost differences between local anesthesia and sedation are inherently different. Surgeon reimbursement in the USA based upon relative value units may incentivize the performance of this procedure in the OR.

It remains to be determined why some providers opt to perform Achilles tenotomy under either partial or general sedation, rather than local anesthesia in the office for patients aged <6 months. This may be attributed partially to provider experience, whereby the younger provider is not as comfortable if the patient can potentially struggle under local anesthesia. Performing tenotomy in the OR was associated with a charge over $4000 greater than in the outpatient clinic [26]. These likely reflect facility fees and other hospital-stay-associated charges, however, the core increase in charge remains substantial with OR tenotomies. Hedrick et al. reviewed charges for 382 patients who underwent a tenotomy in either an outpatient clinic, OR with post-procedure discharge, or OR with post-procedure admission for observation. Further study is needed to determine if this expense is warranted for the typical clubfoot patient presenting shortly after birth and undergoing tenotomy well before the age of six months.

Our study has several limitations, including our respondent rate relative to the overall pool of providers surveyed. Of 1223 individuals surveyed, only 89 indicated they were clubfoot providers and went on to complete the entire survey. Previously, similar surveys sent to POSNA members received over 300 respondents [1,21]. This disparity may be accounted for by “survey fatigue” or the timing of the survey’s delivery in mid-August when many may have been on vacation. An alternative explanation for the low number of self-identified clubfoot practitioners may reflect the shift of clubfoot management towards fewer, more experienced hands. The largest age group answering our survey was those with >30 years’ experience. Additionally, recall bias may exist among respondents when answering survey questions. Our survey was for POSNA membership clubfoot providers only. Therefore, we could not account for patient, family, and socioeconomic factors that may have played a role in the severity of disease from a lack of follow-up [27,28,29,30,31]. Although we captured tenotomy indications and practices, we did not account for who performed the tenotomies in specific institutions. Despite these limitations, this survey provided an updated assessment of how current clubfoot providers practice and treat their patients. This information can be used as a catalyst for the further study of the current trends of clubfoot management. The strengths of the current study come from the ability to capture the current management of clubfoot, specifically casting, tenotomy indications, and anesthesiology practices. There may have been a concentration of subspecialties occurring in pediatric orthopedic groups since the previous studies occurred; this may be evidenced by the predominance of fewer and “grayer” providers in the present study.

## 5. Conclusions

Survey results from the POSNA membership highlight several trends observed in current clubfoot management. More feet are concentrated into fewer, older, and more experienced hands. Providers continue to shift away from using plaster of Paris for serial casting. Further, they are more reliant on partial/general sedation and are more likely to perform tenotomies in the OR. A large percentage of respondents have observed cast saw injuries, indicating a need for the profession to develop and utilize safer methods for cast removal [32].

## Figures and Tables

**Table 1 children-10-00439-t001:** Demographic characteristics of clubfoot providers.

	% of 89 Respondents
**Degree**	
MD	93.3
DO	4.5
PhD	3.4
Other	2.3
**Years in practice**	
<5	14.6
5–10	15.7
10–15	13.5
16–20	14.6
21–25	10.1
26–30	7.9
>30	23.6
**Type of hospital**	
Minor teaching hospital (w/o medical school)	20.2
Major teaching hospital (w/ medical school)	65.2
Federal hospital	2.3
Non-teaching hospital	12.4
**Type of practice**	
University-hospital–based	44.2
Non-university-hospital–based	31.8
Private multi-specialty orthopedic group	5.7
Private pediatric orthopedic group	13.6
Solo private practice	5.7
Pediatric fellowship trained	92.1

DO, doctor of osteopathic medicine; MD, doctor of medicine; PhD, doctor of philosophy; w/, with; w/o, without.

**Table 2 children-10-00439-t002:** Location of practice.

	% of 89 Respondents
Texas	11.9
California	9.5
Canada	6
Ohio	4.8
Washington	4.8
Colorado	3.6
Europe	3.6
Florida	3.6
Hawaii	3.6
Illinois	3.6
New York	3.6
Pennsylvania	3.6
Connecticut	2.4
Maryland	2.4
Missouri	2.4
New Jersey	2.4
North Carolina	2.4
Puerto Rico	2.4
Tennessee	2.4
Virginia	2.4
Other	19

**Table 3 children-10-00439-t003:** Clubfoot management.

	% of 89 Respondents
Management of clubfoot has changed throughout training	57.5
Participation in a clubfoot training course	51.7
Directly from Dr. Ponseti (Iowa)	36.4
Iowa (after Dr. Ponseti)	27.3
International Pediatric Orthopedic Symposium	59.1
Baltimore Limb Deformity Course	31.8
Other	9.1
**Casts used**	
<4	8.1
4–7	88.7
Other	3.2
**Days cast in place**	
4	3.3
5	0
6	1.6
7	93.4
>7	1.6
**Type of cast**	
Long leg cast	98.4
Short leg cast	1.6
Material	
Plaster	69.4
Fiberglass	3.2
Semi-rigid	27.4
**Type of brace**	
Denis Browne	60.8
Mitchell–Ponseti	72.9
Dobbs	31.7
Cunningham	2.4
ADM brace	4.7
AFO	20.1
KAFO	8.2
Bebax shoe	1.7
Patients requiring tenotomy	84.8
**Type of anesthesia**	
Cold spray	1.2
Local injection	39.5
Numbing gel	43
Conscious sedation	7
General	33.7
None	5.8
Other	7
**Tenotomy location**	
Operating room	46.5
Clinic/office	45.4
Procedure room	8.1
**Tenotomy age indication**	
<6 months	63.1
6–9 months	10.7
9–12 months	7.1
1–2 years	1.2
Other	28.6

ADM, abduction dorsiflexion mechanism; AFO, ankle foot orthosis; KAFO, knee ankle foot orthosis.

## Data Availability

The data presented in this study are available upon request from the corresponding author.

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
