# Peer review of "Current Clubfoot Practices: POSNA Membership Survey"

_children, 2023, doi:10.3390/children10030439_

Round 1

Reviewer 1 Report

I would like to congratulate the authors on a well-written manuscript. However, I feel that there is a scope for further improvements in the manuscript. I would suggest the authors make the following improvements: 

1. There is a need to investigate the relationship between various aspects of treatment practices with physicians' experience and geography using statistical analysis. 

2. The latest articles on clubfoot treatment are missing from citations. There is a need to map the findings of this study with the latest research.  The author may find the following related papers interesting  (https://doi.org/10.1016/j.jor.2023.01.009, https://doi.org/10.4103%2Fjfmpc.jfmpc_2606_20, https://doi.org/10.1007/s43465-022-00748-8,   DOI: 10.1097/JPO.0000000000000431, https://doi.org/10.1016/j.eclinm.2022.101448, https://doi.org/10.1016/B978-0-323-91911-1.00012-2). 

Author Response

  1. With tracked changes turned on, we have inserted at line 235 (amidst our paragraph on limitations) the following: "Our survey was for POSNA membership clubfoot providers only. Therefore, we could not account for patient, family, and socioeconomic factors that may have played a role in severity of disease from lack of follow-up [28-32]. Although we captured tenotomy indications and practices, we did not account for who performed the tenotomies in specific institutions. Despite these limitations, this survey provided an updated assessment of how current clubfoot providers practice and treat their patients. This information can be used as a catalyst for further study of the current trends of clubfoot management. "
  2. These articles are now cited (at the above added text) and included in the reference list: 
    -(28) Singh, S.; Mali, H.S.; Jain, A.K. Contemporary challenges in clubfoot treatment: A quantitative study among Indian parents. J Orthop 2023, 37, 5-8.
    -(29) Iqbal, M.S.; Dubey, R.; Thakur, K.; et al. Assessment of awareness and barriers to clubfoot treatment in the Indian scenario. Fam Med Prim Care Rev 2021, 10, 4229-4235.
    -(30) Pinto, D.; Leo, D.G.; Aroojis, A.; et al. The impact of living with clubfoot on children and their families: Perspectives from two cultural environments. Indian J Orthop 2022, 56, 2193-2201.
    -(31) Pigeolet, M.; Vital, A.; Daoud, H.A.; et al. The impact of socio-economic factors on parental non-adherence to the Ponseti protocol for clubfoot treatment in low- and middle-income countries: A scoping review. EClinicalMedicine 2022, 48, 101448.
    -(32) Singh, S.; Mali, H.S. Foot abduction orthosis compliance in clubfoot treatment: biomechanics, challenges, and future scope. In: Sandhu, K.; Singh, S.; Prakash, C.; Subburaj, K.; Ramakrishna, S.; eds. 3D Printing in Podiatric Medicine. Amsterdam, Netherlands: Elsevier; 2023, 103-121.

Reviewer 2 Report

This manuscript aimed to identify current treatment practices among clubfoot practitioners within the Pediatric Orthopaedic Society of North America (POSNA). A 23-question online survey of members was conducted between June and August 2021. Eighty-nine respondents self-identified as clubfoot providers. Of these, 93.1% had an MD degree, 23.6% possessed > 30 years’ experience, and the majority (65.6%) worked in a teaching hospital associated with a medical school. Most responders (92.0%) were pediatric fellowship trained. 51.7% had participated in a clubfoot training course.

I read the article with interest, the title is well thought out and faithfully reflects the content of the study.

The abstract is very useful to frame the purpose of the study.

In the introduction, the characteristics of Clubfoot have been described.  The materials and methods have been adequately development. The discussion is sufficiently described.

Nevertheless, some minor changes are needed to be considered suitable for publication.

Comment 1: In the introduction: Some information about etiology, diagnosis and treatment of clubfoot should be deepened please adding appropriate bibliographical references. (Pavone V. et al (2022) " Early developmental milestones in patients with idiopathic clubfoot treated by Ponseti method").

Comment 2: In the material and methods: Please, add more information about the experience of operators analyzing patient data, were all orthopedic surgeons? What kind of experience did they have with pathology?

Comment 3: In the material and method: The tenotomies were performed in the same institute and by the same surgeon?

Comment 4: In the material and methods: What kind of classification did you use to estimate the severity of the clubfoot?

Comment 5: In the discussion: It would be better referring to previous studies performed on the same topic, for example (Pavone V. et al (2021) "Sport Ability during Walking Age in Clubfoot-Affected Children after Ponseti Method: A Case-Series Study ").

Comment 6: In the discussion: It would be advisable to clearly refer to the limitations of the study

Comment 7: Finally, additional English editing is needed. The Non-Native Speakers of English Editing Certificate was not signed.

Author Response

  1. We have added text to the introduction including the Pavone reference as you suggest. With tracked changes turned on, starting at line 55, please find this new text: "Clubfoot is an idiopathic developmental deformation characterized by excessive shortening of the tibialis posterior tendon resulting in the classic “CAVE” deformity (Cavus; forefoot Adductus; hindfoot Valgus; and Equinus contracture). It represents one of the most prevalent congenital birth defects with an estimated 1 per 1000 live births affected [16]. Although poorly understood, the etiology of clubfoot is thought to be contributed primarily by intra-uterine posture, with scant literature suggesting alternatives related to musculoskeletal, neurologic, or vasculature mechanisms [17]. Environmental and genetics have been posited to contribute to the etiology of clubfoot, specifically with the expression of both PITX1 and TBX4 pathways and their role in hindlimb development [18]. Regardless of etiology, clubfoot has successfully been managed for several decades. Despite the method’s success, patients treated using the Ponseti technique have been shown to achieve delayed independent gait [19]. However, this may not be surprising considering the initial deformity that impedes the child’s ability to ambulate."
  2. At line 93 we have clarified: "Members are pediatric orthopedic-trained practitioners and allied health professionals" and at line 115 "nearly all respondents were pediatric fellowship trained orthopedic surgeons"
  3. In our paragraph about limitations, at line 237, we have added "Although we captured tenotomy indications and practices, we did not account for who performed the tenotomies in specific institutions."
  4. Thanks for your comment. There are several classification systems used for clubfoot management, but we did not ask what the preferred classification used for each provider. We thought that this would be out of the scope of the survey we provided.
  5. Thanks for this comment. We added the other recommended studies that were pertinent to our paper, however this paper, we felt, was outside the scope of our paper. We believe that the inclusion of other relevant publications were included.
  6. Our limitations are listed throughout the paragraph that begins at line 226. We have added additional text to this paragraph as well at new lines 235-242.
  7. We respectfully disagree that further English editing is required. We are not aware of a requirement to provide any such certificate. This article was edited by a professional medical editor (Robert Farley, thanked in the acknowlegments).

Reviewer 3 Report

Dear Author,

Thank you for the opportunity to review your article.

It is interesting to find that only a small fraction of pediatric orthopedic surgeons provide clubfoot treatment in the US.

The introduction is elaborate, the Materials and Methods are succinct, and the results are properly illustrated.

Our experience tends to be similar regarding OR tenotomies nowadays, but every pediatric surgeon treats this pathology, not only the old and experienced. Plaster of Paris is definitely better for their small feet because of the possibility to mold properly and the lower risk of complications.

Author Response

Thank you for your review and comments. It is not our intention to allude that only the old and more experienced surgeons treat clubfoot, and we are not attempting to infer what material is superior; we are merely attempting to report the findings of our survey. We hope this is clear in lines 249-251 of our conclusion. 

Round 2

Reviewer 1 Report

The article now has sufficient merit for publication.